# Real-Life Study of the Benefit of Concomitant Radiotherapy with Cemiplimab in Advanced Cutaneous Squamous Cell Carcinoma (cSCC): A Retrospective Cohort Study

**DOI:** 10.3390/cancers15020495

**Published:** 2023-01-13

**Authors:** Barbara Bailly-Caillé, Diane Kottler, Rémy Morello, Marie Lecornu, William Kao, Emmanuel Meyer, Anne Dompmartin, Jean-Matthieu L’Orphelin

**Affiliations:** 1Department of Dermatology, Caen-Normandie University Hospital, 14000 Caen, France; 2Biostatistics and Clinical Research Unit, Caen-Normandy University Hospital, 14000 Caen, France; 3Department of Radiotherapy, Francois Baclesse Center, 14000 Caen, France; 4Department of Radiotherapy, Cotentin Public Hospital, 50100 Cherbourg-en-Cotentin, France; 5Department of Radiotherapy, Maurice Tubiana Center, 14000 Caen, France

**Keywords:** cutaneous squamous cell carcinoma, immunotherapy, cemiplimab, concomitance, radiotherapy, abscopal effect, adverse event

## Abstract

**Simple Summary:**

Cemiplimab is used in the treatment of advanced cutaneous squamous cell carcinoma. Synergistic antitumoral response by concurrent radiotherapy and immunotherapy seems promising. We collected data from patients who were administered cemiplimab in Caen Hospital. Our primary objective was to determine the best overall response and overall survival rate. The secondary objectives were the disease control rate, median time to response, progression-free survival, overall survival, clinical benefit of radiotherapy, and safety data. The objective response rate was 45.5%, including 47.6% in the cemiplimab group and 41.6% in the concomitant group. Concurrent radiotherapy enables an earlier clinico-radiological response, and the response to treatment was prolonged despite discontinuation of cemiplimab. Radiation therapy also provided a therapeutic effect without increasing the occurrence of adverse events. Our real-life study confirms the efficacy of cemiplimab in the treatment of advanced cSCC and the benefit of concurrent radiotherapy in achieving a quicker and persistent clinico-radiological response, without any safety alert.

**Abstract:**

Background: Cemiplimab is a monoclonal antibody targeting the PD-1, and phase II trials have shown its efficacy in the treatment of advanced cutaneous squamous cell carcinoma in patients who are not candidates for curative surgery or radiation therapy as a first- or later-line treatment. A synergistic antitumoral response has been demonstrated with concurrent radiotherapy and PD1-immunotherapy. However, no real-life study has demonstrated this effect in advanced cutaneous squamous cell carcinoma. Methods: We conducted a real-life retrospective cohort study to investigate the benefit of concomitant therapy in 33 patients treated with cemiplimab at the University Hospital of Caen, alone (C group) or concomitant to radiotherapy (C/RT group). Our primary objectives were to evaluate the best overall response and objective response rate. Our secondary objectives were the disease control rate, median time to response, progression-free survival, overall survival, clinical benefit of radiotherapy, and safety data. After stopping cemiplimab administration, we performed a follow-up of our patients and performed a descriptive study. Results: We reported an objective response rate of 45.5%, including 47.6% in the cemiplimab group versus 41.6% in the concomitant group. The addition of radiotherapy to cemiplimab enables an earlier clinico-radiological response, with a median duration of 5.5 months in the cemiplimab group versus 3 months in the concomitant therapy group. The response to treatment was prolonged despite discontinuation of cemiplimab, with 91.6% (*n* = 11/12) and 83.3% (*n* = 10/12) patients in complete or partial remission at 6 months and 1 year after cessation of cemiplimab and no switch to another oncological treatment, respectively. Radiation therapy also provided a therapeutic effect in 83.3% of the patients in the concomitant group, without increasing the occurrence of adverse events. Conclusions: Our study confirms the efficacy of cemiplimab and radiotherapy in the management of advanced cutaneous squamous cell carcinoma. Concomitant therapy permitted to obtain an earlier radiological response, a beneficial local therapeutic effect of radiotherapy, without any safety alert.

## 1. Introduction

Cutaneous squamous cell carcinoma (cSCC) is the second most common skin cancer, with an estimated incidence of 30/100,000 inhabitants. The prognosis is often excellent, with a 5-year survival rate of over 90%. Advanced cSCC includes locally advanced cSCC that cannot be treated with surgery or radiation therapy, or metastatic locally or distantly cSCC [1] and advanced cSCC represents 2–5% of cSCC.

Immune checkpoint inhibitors (ICIs), especially anti-programmed cell death-1 receptor (anti-PD1) antibodies such as pembrolizumab and cemiplimab, have shown improvement in outcomes and have recently been approved for the treatment of advanced cSCC [2,3]. The response to immunotherapy is correlated with the tumoral mutation burden, which is very high in cSCC [4,5,6].

Several molecules have been studied, including pembrolizumab and cemiplimab, which are now part of the therapeutic arsenal for advanced cSCC [7,8]. However, immunotherapy alone does not provide a consistent response, as some patients do not respond. For those with a clinical response, the response is not always durable [9].

Almost half of all cancer patients will receive radiation therapy [10]. Radiation therapy is often used in cSCC as adjuvant therapy in cases of poor prognosis factors to reduce the risk of local relapse but is increasingly used as curative therapy when surgery is not possible and for palliative purposes to improve patient quality of life by decreasing pain, bleeding, and tumour size [11].

Radiotherapy has a direct effect on tumoral cells because it induces cellular DNA damage. It also has an immunogenic effect, with immunogenic cell death induced by three important signals: calreticulin, HGMP-1 and ATP [12]. Radiotherapy not only induces new immunity by generating neoantigens but also stimulates existing immunity by its action on the tumour microenvironment [13]. The immunogenic effect of radiotherapy was especially observed at hypofractionated doses [14], which could allow for responses in distant nonirradiated sites, called the abscopal effect [15].

Concomitant radiotherapy with immunotherapy is a very promising method of treatment due to a synergistic antitumour response that occurs by increasing the systemic immune response [16,17]. This synergistic effect has been demonstrated in numerous clinical studies in melanoma [18,19], non-small cell lung cancer [20], and in head-and-neck squamous cell carcinoma [21]. To our knowledge, no real-life studies have demonstrated this effect in advanced cutaneous squamous cell carcinoma.

This real-life retrospective study aims to investigate the benefit of concomitant radiotherapy with cemiplimab in the management of advanced cutaneous squamous cell carcinoma. 

## 2. Materials and Methods

We performed a retrospective single-centre cohort study in Caen Hospital. Data from patients receiving cemiplimab for advanced cutaneous squamous cell carcinoma (*n* = 33) were collected from 1 October 2018 to 31 August 2021. There were no exclusion criteria. Patients received intravenous cemiplimab until death, palliative care decision, toxicity or discontinuation due to the loss of reimbursement for cemiplimab in August 2021 in Caen Hospital. The patients were categorized into those who received cemiplimab alone (C group) and those who had concurrent cemiplimab and local radiotherapy (C/RT group). The choice of the type of treatment was validated in a multidisciplinary meeting for all patients, according to the progress of the disease, the site of the metastases, the medical history of the patients and in particular the history of radiotherapy, the absence of contraindications to radiotherapy, and the general state of the patient. We performed a second time a complementary medical data collection for 1 year after discontinuation of cemiplimab, according to a descriptive approach from 1 September 2021 to 1 September 2022. 

We recorded data relating to the general characteristics of the patients (age, sex, ECOG status), medical history (previous history of squamous cell carcinoma, immune status, presence of immunosuppression or lymphopenia), clinical or histological risk factors (tumour location and size, perineural invasion, invasion beyond subcutaneous fat), and previous treatment (lines of therapy or adjuvant radiotherapy). For patients who have received concomitant radiotherapy, complementary medical data were collected (intent of radiotherapy: palliative or curative, site of radiotherapy, dose per fraction, fraction and biologically effective dose (BED)) in the radiotherapy centre. BED was calculated using an alpha/beta ratio of 10. Palliative radiotherapy was used to improve patient quality of life by decreasing pain, bleeding, and tumour size without curative objectives.

The study complied with the ethical standards resulting from the Declaration of Helsinki and was approved by the ethics committee of the University Hospital of Caen. This observational study did not involve the patients in any way different from their usual care (reuse of their health data). An information sheet was given to each patient to collect their nonopposition to the study, and none of them were opposed to it.

All patients were re-evaluated as follows:-Clinically every 2 or 3 weeks before the administration of cemiplimab;-Radiologically every 3 months with TEP tomodensitometry, cerebro facial MRI or thoraco abdomino pelvic scanner according to tumour localization, using the iRECIST criteria.

A patient was considered to have complete response (CR) when the response was both clinical and radiological according to the iRECIST criteria.

The primary endpoints evaluated were the best overall response rate (BOR) and objective response rate (ORR), defined by the presence of complete (CR) or partial (PR) response. The secondary endpoints were the disease control rate (DCR), time to response, progression-free survival (PFS), overall survival (OS) and safety. For the patients who received concomitant radiotherapy, local improvements and positive clinical effects were also evaluated (size reduction, reduction of bleeding or pain). The adverse events (AEs) of cemiplimab were graded from 1 to 5 according to the Common Terminology Criteria for Adverse Event (CTCAE version 4) and for radiation therapy according to the National Cancer Institute- Common Toxicity Criteria (NCI-CTC).

DCR was defined by the presence of a stable, partial or complete response for more than 12 weeks; OS was defined by the time from the beginning of cemiplimab to death or palliative care decision (OS) or progression (PFS).

### Statistical Analysis

Data were expressed as the mean ± standard deviation or median (range: Q1–Q3)) and percentage, depending on the variable of interest. Baseline data were compared between the patients on cemiplimab with and without radiotherapy using chi-square or Fisher’s exact tests for qualitative variables and using an analysis of variance or the Mann–Whitney test for continuous variables. A Kaplan–Meier survival analysis was performed to test survival differences between the patients on cemiplimab with and without radiotherapy. The distributions of survival times were compared using the log-rank test, and Kaplan–Meier curves were generated. Univariate and multivariate Cox models were used to assess the prognosis during follow-up between the deceased/palliative care groups versus the surviving/routine care groups on the one hand, and between the groups with or without progression on the other hand. The results of the multivariate Cox models were obtained after eliminating stepwise covariates with descending significance level (*p*-val-ue). The selection of categorical and continuous variables (as independent variables) that were included in the model was based on a significance threshold equal to 0.10 in the univariate Cox model. The adjusted hazard ratios (HRs) are provided with 95% confidence intervals (95% CIs). In addition, the receiver operating characteristic (ROC) curves were plotted, and the area under the curve was computed (c-index). We selected the point on the ROC curve that maximized the sensitivity and specificity. We used IBM-SPSS 23.0 software (Armonk, NY, USA: IBM Corp.) for data analysis. All tests were 2-sided, and a *p*-value < 0.05 was considered statistically significant.

## 3. Results

### 3.1. Patients

We included 33 patients who received cemiplimab at Caen University Hospital from October 2018 to August 2021. The cohort was comprised of 21 patients who received cemiplimab alone (C group) and 12 patients who received concomitant radiotherapy and cemiplimab (C/RT group).

The baselines’ patient characteristics are reported in Table 1. The general characteristics, medical history, clinical and histological risk factors were similar between the two groups. A majority of the patients with advanced cSCC had metastatic disease (85.7% in the C group and 91.7% in the C/RT group) rather than locally advanced cSCC. The only statistically significant difference between the two groups was the history of previous radiotherapy, which was more frequent in the C group. Five patients received two sequences of radiotherapy. Of the patients who received concurrent cemiplimab with radiotherapy, the mean dose per fraction was 4.0 ± 1.7 Gy in 16.2 ± 12.6 fractions.

Detailed data on baseline characteristics for each patient are available in the Appendix A.

### 3.2. Efficacy Evaluation during Cemiplimab Treatment

The objective response rate was 45.5% (95% CI [28.1–63.7] %) with 15/33 responders and 47.6% (*n* = 10/21) (95% CI [25.7–70.2] %) and 41.6% (*n* = 5/12) (95% CI [15.2–72.3] %) in C and C/RT groups, respectively (*p* = 1.000). 

There were 10/33 (30.3%; 95% CI [15.6–48.7] %) complete responses (CR): 6/21 (28.6%; 95% CI [11.3–52.2] %) in the C group and 4/12 (33.3%; 95% CI [9.9–65.1] %) in the C/RT group. There were 5/33 (15.2%; 95% CI [5.1–31.9] %) partial responses (PR) with 4/21 (19.1%; 95% CI [5.5–41.9] %) patients in the C group and 1/12 (8.3%; 95% CI [0.2–38.5] %) patient in the C/RT group, with no significant difference (*p* = 0.229).

Among the 10 complete responders, none relapsed during the follow-up, and one patient was lost to follow-up. Only two PR patients had a progressive disease during the follow-up. In contrast, only one of the five SD (stable disease) patients had a durable response without any relapse.

Among the patients with reported progression (*n* = 12/33): six progressions were immediate, four were after initial SD and two were after initial PR. Considering disease progression, we reported no significant difference between the C and C/RT groups regarding local or distant progression.

Seven patients had no radiological evaluation because of death (*n* = 5), palliative care decision (*n* = 1) or loss of follow-up (*n* = 1). 

The disease control rate was 70.0% (95% CI [45.7–88.1] %) with 81.8% (95% CI [48.2–97.7] %) and 55.6% (95% CI [21.2–86.3] %) in the C and C/RT groups, respectively (Table 2). 

Detailed data on results for each patient are available in the Appendix A.

The median follow-up was 7.0 [2.5–20.5] months in the C group and 8.5 [7.0–10.0] months in the C/RT group (*p* = 0.782), with a median treatment duration of 7.0 [2.0–20.5] months in the C group and 7.0 [4.5–9.5] months in the C/RT group (*p* = 1.000). 

The median time to response was shorter and numerically significant in the C/RT group: 3.0 [2.5–3.5] months vs. 5.5 [3.0–14.3] in the C group (*p* = 0.075). 

The median PFS was 6 months in the C/RT group and was not reached in the C group. The Kaplan–Meier estimated that the 6-months PFS was 50% in the C/RT group and 77% in the C group. The median OS was 9 months in the 2 groups, and the Kaplan–Meier estimated 6-months OS was 55% in the C group and 92% in the C/RT group (Figure 1).

Radiotherapy showed a positive clinical effect in 83.3% of the patients (*n* = 10/12). A comparison test against the hypothesis of no effectiveness (50/50) found a therapeutic effect of radiotherapy (*p* = 0.037). Median OS was not reached when radiotherapy was used with curative intent, versus 7 months when the intent was palliative (Figure 2).

Prior therapeutic line and number of cemiplimab administrations were risk factors for progression and palliative care decision or death in the univariate analysis. Age and tumour size were associated with progression and the intent of radiotherapy with the palliative care decision or death. The multivariate analysis highlighted that the number of cemiplimab administrations (<10 versus ≥10) was associated both quantitatively and qualitatively with significantly better overall survival and progression-free survival after adjustment for cofounding variables (Table 3).

The best balance point was ten sessions of cemiplimab, which relied on the sensitivity and specificity in the ROC curves representing the probability of no palliative care decision or death according to the number of cemiplimab administrations (Figure 3 and Figure 4). The prognostic parameters were as follows: sensitivity = 88.2% (95% CI [63.6–98.5] %); specificity = 87.5% (95% CI [61.7–98.5] %); positive predictive value = 88.2% (95% CI [67.0–96.5] %), negative predictive value = 87.5% (95% CI [65.3–96.3] %); positive likelihood ratio = 7.06 (95% CI [1.9–26.1]); negative likelihood ratio = 0.13 (95% CI [0.04–0.5]). 

We studied the evolution and the overall survival according to the age. There was no significant difference, as illustrated in Figure 5. 

### 3.3. Analysis after Discontinuation of Cemiplimab

Cemiplimab was stopped in August 2021 because of the loss of temporary authorization for the use of cemiplimab by the French transparency commission. At this point, we just performed a follow-up of survivor patients (*n* = 14) without introducing any other oncological drug; 12 were still responding, with 9 CR and 3 PR. Only one patient was still stable, and one patient had progression. 

At 6 months, we reported 2 deaths and 1 loss to follow-up. At 1 year of discontinuation of cemiplimab, we reported *n* = 8 patients with CR, *n* = 2 patients with SD and *n* = 1 patient with progressive disease (PD), (Figure 6). 

### 3.4. Safety

We reported no safety alert. Of the 21 patients in the C group, we reported no adverse events in *n* = 10 (47.6%), and *n* = 5 (23.8%) and *n* = 6 (28.6%) had grade III and grade I–II AEs, respectively. For the 12 patients in the C/RT group, we reported no adverse events due to cemiplimab in *n* = 5 (41.7%), with *n* = 3 (25.0%) and *n* = 4 (33.4%) grade III and grade I–II AEs, respectively. We reported no deaths due to cemiplimab.

We reported no adverse effects in 50.0% (*n* = 11) of the patients over 75 years old compared to 36.4% (*n* = 4) of those under 75 years of age.

The median time between the introduction of cemiplimab and the first adverse event was 7 months, with 7 and 3 months in the C and C/RT groups, respectively (*p* = 0.425). We reported a statistically significant difference in the median time to occurrence of grade I–II AEs, with a median time of 10 months in the C group and 3.5 months in the C/RT group (*p* = 0.019). Eight of the twelve patients who received concomitant radiotherapy had AEs due to radiotherapy, and all these AEs were grade I or II.

## 4. Discussion

To our knowledge, this is the first real-life retrospective study to investigate concomitant radiotherapy to cemiplimab underlying that the abscopal effect in cutaneous squamous cell carcinoma could be a significant therapeutic strategy. We report a quicker favourable response to cemiplimab with a median time to response of 3 months in the concomitant group versus 5.5 months in the cemiplimab group. Concomitant radiotherapy adds a clinical effect in 83.3% of patients, although there is no radiological response. Furthermore, the safety profile of this concomitance appears acceptable. We report an important objective rate response with long responders, even after stopping cemiplimab (*n* = 11/12 at 6 months and *n* = 10/12 at 1 year, i.e., 91.6% and 83.3%, respectively). Our results confirm the place of cemiplimab in the management of cSCC, including patients with severe comorbidities.

The characteristics of the patients were similar in both groups (C group versus C/RT group), with a median age and sex ratio similar to previous real-life studies [22,23,24]. The unique significant difference was prior radiotherapy in 14/21 of patients in the C group and one patient in the C/RT group (Table 1). Limb localization was found in 25% of the patients in the C/RT group, mainly including men, and this was a result superior to the German study (9%) [22]).

The objective response rate in our study was 45.5%, which is similar to the results from other trials evaluating cemiplimab (47% in the phase I–II study [7], 50% in the French real-life study [23], 58% in the Italian real-life study [24] and 58.7% in the German real-life study [22]). The factors that could explain the slight difference would be the higher proportion of patients with ECOG status > 2 in our study (45.5% versus 27% in French [23] and 17% in Italian [24] studies). Moreover, ECOG status > 2 was an exclusion criterion in the phase I–II study [7]. In addition, our study included 29/33 patients with metastatic disease (87.9% versus 65% in the French [23], 30.5% in the Italian [24] and 87% in the German [22] studies). Only 12.1% of the patients presented with a locally advanced cSCC.

Concomitant radiotherapy to cemiplimab does not seem to improve the objective response rate, which is similar to the results that were previously published for metastatic head and neck squamous cell carcinoma treated with nivolumab and radiotherapy [25]. Data from the literature support a more immunogenic effect of radiotherapy with a hypofractionated regimen (between 3 and 5 fractions of 6 Gy). This immunogenic effect is based on immunogenic cell death and is also based on promoting innate and adaptative immune responses [14]. Therefore, we assume that the radiotherapy delivery was suboptimal for some of our patients: seven patients received a hypofractionated regimen between 2 and 6 fractions of 6 Gy on at least one of the radiotherapy sequences, and 5 others received a low-dose immunomodulatory regimen between 15 and 35 fractions in 1.8 to 4 Gy. None of the patients with limb cSCC (3 in the C/RT group and 1 in the C group) responded to cemiplimab. These results could be explained by a lower tumour mutational burden in these sun-protected areas, as the anti-PD1 response is correlated with tumour mutational burden [4]. 

We reported a disease control rate of 70%, with no significant difference between our two groups (81.8% in the C group versus 55.6% in the C/RT group). This result was superior to that of real-life studies of cemiplimab (59.6%, 71.1% and 80.4% in French [23], Italian [24] and German [22] real-life studies, respectively). This result confirms the place of cemiplimab in the management of advanced cSCC with a strong antitumour response, and a long-term response.

The median follow-up time and treatment were comparable to those in the literature. We can observe the presence of wide confidence interval in C group. This result suggests more homogeneity in patients of the C/RT group with a similar evolution and response. It could be explained by a small cognitive bias in the multidisciplinary meetings. Our study reports a time reduction of 2.5 months to response with concurrent radiotherapy and cemiplimab (the median time to response was 5.5 months in the C group versus 3 months in the C/RT group). This promising result could be explained by a synergistic effect of concomitant radiotherapy and immunotherapy. Cemiplimab reinforces the existing immunological response against tumour cells. Radiotherapy creates neoimmunity due to the liberation of tumoral antigens and strengthens responses by acting on the tumoral environment. This synergistic effect stimulates the immunological response against tumours, promoting T-cell activation and enhancing T-cell activity, which is also called a abscopal effect when it appears on nonradiated sites [26].

Cemiplimab seems to induce a long-term persistent response. Indeed, none of the patients in complete response relapsed, and 3 of the 5 patients with partial response were still responders during the follow-up. This suggests a persistence of the immunogenic effect of cemiplimab even after treatment cessation. When cemiplimab was stopped, only one PR patient relapsed at 6 months and one CR patient died during follow-up, whose death was unrelated to the oncological disease. To avoid progression after discontinuation of cemiplimab, we suggest cemiplimab administration spacing in long responder patients, similar to that proposed in melanoma and Merkel’s carcinoma [27].

The OS at 6 months was 92% in the C/RT group and 55% in the C group. The median OS was 9 months in both groups. In real-life studies of cemiplimab [22,23,24], the overall survival was not reached. We probably underestimated the overall survival for 6 patients because of the lack of information on the date of their death, which was assimilated to the date of palliative care decision. Progression-free survival was less relevant in our advanced-disease population. 

Concomitant radiotherapy has a real initial beneficial effect because it reduces the duration to obtain an oncological response and improves the 6-months OS. The responses do not seem to be persistent over time in terms of overall survival (median overall survival at 9 months in both groups). It is known by onco-dermatologists that the carcinological response can take several months to arrive after the introduction of immunotherapy. In the face of aggressive and rapidly evolving diseases or debilitated terrain (i.e., profoundly immunocompromised patients), it is not always easy to wait 3 to 6 months to judge the clinical and radiological response. In this sense, even if the addition of radiotherapy to cemiplimab does not change the median OS, it enables a quicker response and better containment of certain cutaneous squamous cell carcinoma that could not be controlled with immunotherapy alone. The Kaplan–Meier analysis of OS showed a very significant difference in overall survival for curative radiotherapy compared to palliative radiotherapy (Figure 2). The degree of significance was very low (*p* < 0.001), despite the small sample size and confounding factors. This reinforces the strengths of these results. We highlighted the benefit of concurrent radiotherapy to cemiplimab, suggesting at least a local immune effect, but no abscopal effect was observed.

In our study, 14 patients in the C group could not receive radiotherapy because of previous radiotherapy with a cumulative dose. Neoadjuvant cemiplimab was proposed in the ASCO 2022 [28], allowing long-term therapeutic response and adjuvant radiotherapy savings. Neoadjuvant cemiplimab could be administered with a good tumoral response, and it would enable radiotherapy dose savings with a possible new line of treatment with concurrent radiotherapy and cemiplimab in curative situations. This new management of carcinomas with poor prognoses should be discussed in interdisciplinary oncology groups.

The occurrence of adverse events was comparable to that in other studies. Severe grade III–IV AEs were slightly higher in real-life studies (24.3% with 23.8 and 25% in the C and C/RT groups, respectively, versus 9% in the French study [23], 10.7% in the Italian study [24], and 13% in the German [22] study). However, we collected all AEs, with a probable overestimation of the gradation of AEs because of the absence of exclusion criteria and a higher proportion of patients with an ECOG status >2. No increase in toxicity was observed with concomitant therapy. However, grade I and II adverse events appeared more rapidly. This result could be explained by the same mechanism leading to a faster radiological response. 

Most of patients who were over 75 years old (66.7%) had a better OS on Kaplan–Meier analysis. Furthermore, the occurrence of AEs was lower. Immunotherapy seems to be effective and safe for patients over 80 years of age [29], despite the immune senescence and immune system deregulation that affect T cells in the geriatric population [30]. Older patients may have a better response, with a better tolerance than younger patients [31,32].

Despite the retrospective and monocentric nature, and small sample size, our study reinforces interest in concomitant radiotherapy to cemiplimab in the management of advanced cSCC. This study does not have sufficient numbers of subjects to make practice recommendations. However, our results are noteworthy because, as far as we know, this is the first study to address this issue and the value of concurrent immunotherapy and radiotherapy in sSCC. It could serve as a basis for designing future studies by prospectively comparing the different carcinological responses between cemiplimab and radiotherapy concomitant to cemiplimab, with similar populations, larger numbers and similar diseases in order to have fully comparable groups. It also highlights the need for cooperation between onco-dermatologists, medical oncologists and radiotherapists to establish a coherent care pathway and increase the therapeutic solutions available.

## 5. Conclusions

Our study confirms the efficacy of cemiplimab in the treatment of advanced cSCC, in accordance with various recent real-life studies. Moreover, the clinico-radiological response to cemiplimab seems to be long-term and persistent even one year after stopping any oncological treatment. This is the first real-life study dedicated to evaluating concomitant radiotherapy to cemiplimab. This combination of cemiplimab and radiotherapy allows for a quicker objective clinico-radiological response than with cemiplimab in monotherapy, as well as an improvement in local symptomatology, without increasing AE occurrence. Thus, the combination of radiotherapy with cemiplimab is very promising in the treatment of advanced cSCC. Although this is not a clinical trial and therefore results should be interpreted with caution, these data are relevant for daily practice as well as a basis for designing future trials. Further prospective and randomized studies are needed to investigate this concomitance, including neoadjuvant and adjuvant treatment.

## Figures and Tables

**Figure 1 cancers-15-00495-f001:**
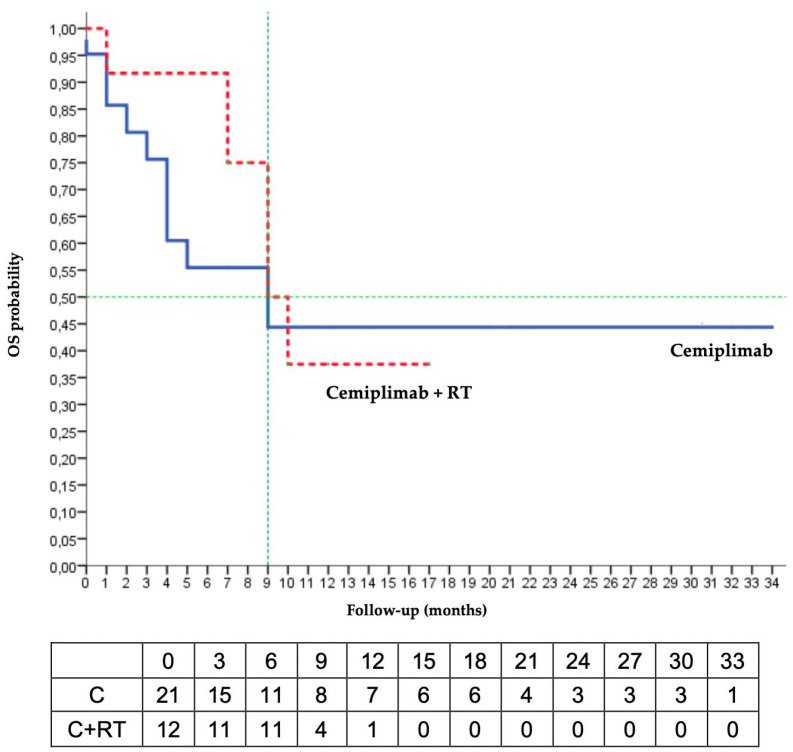
Kaplan–Meier analysis of overall survival (RT radiotherapy; OS: overall survival).

**Figure 2 cancers-15-00495-f002:**
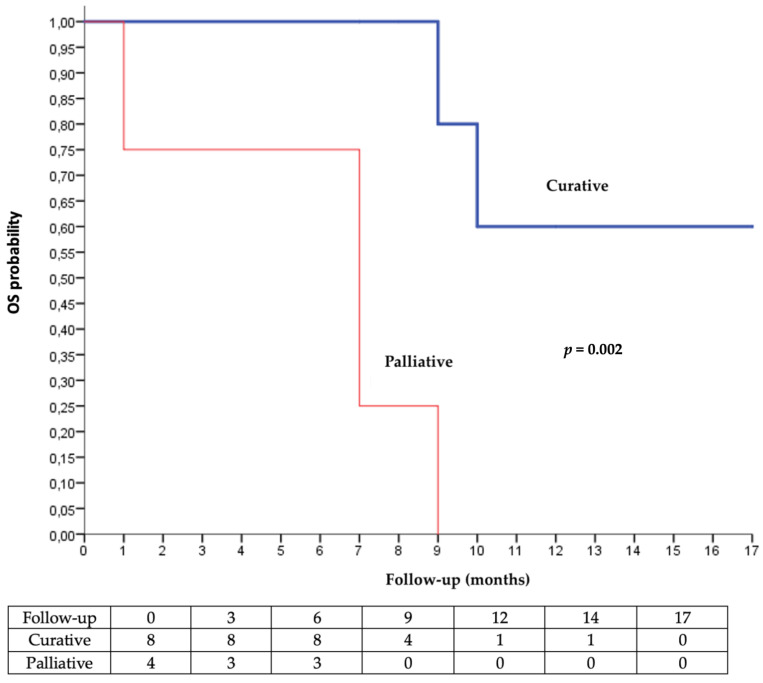
Kaplan–Meier analysis of overall survival according to the intent of radiotherapy (palliative vs. curative).

**Figure 3 cancers-15-00495-f003:**
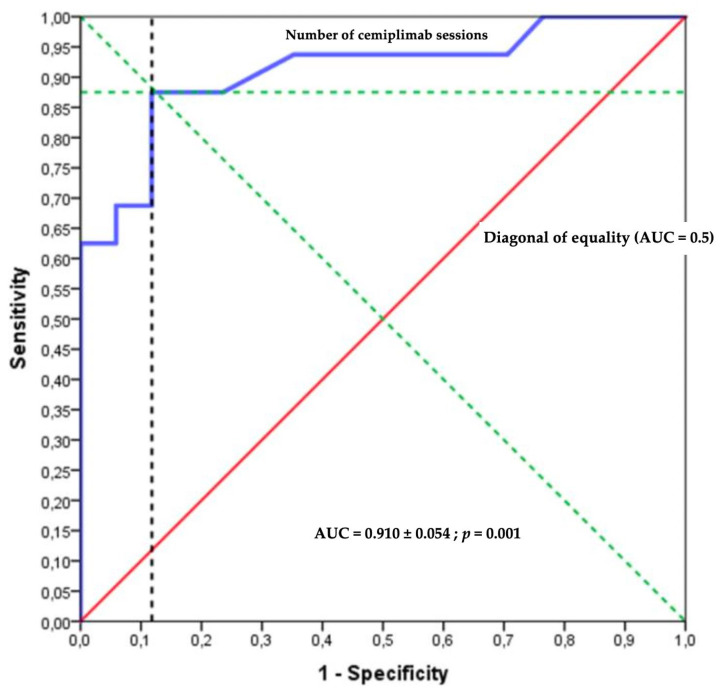
Receiver operating characteristic (ROC) curve.

**Figure 4 cancers-15-00495-f004:**
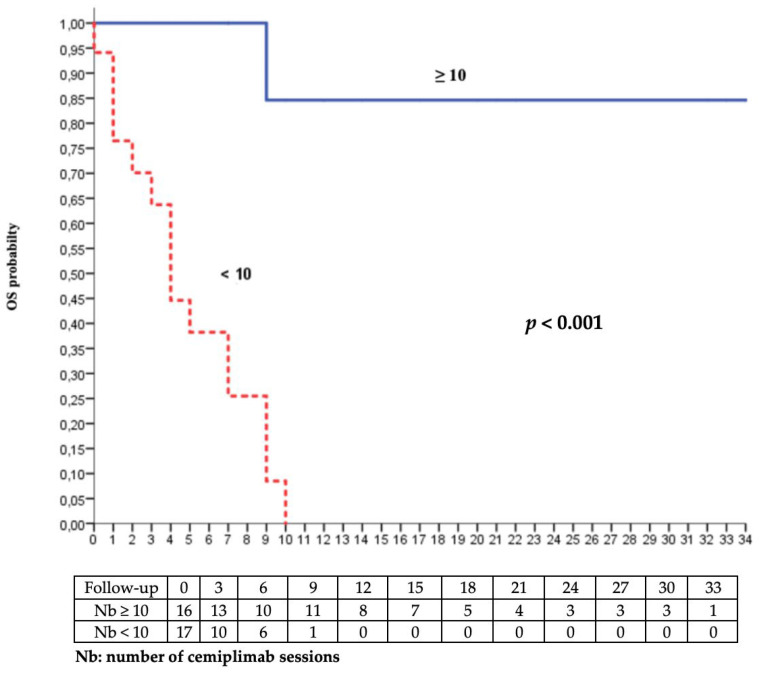
Kaplan–Meier analysis of overall survival according to the number of cemiplimab sessions (<10 vs. ≥10).

**Figure 5 cancers-15-00495-f005:**
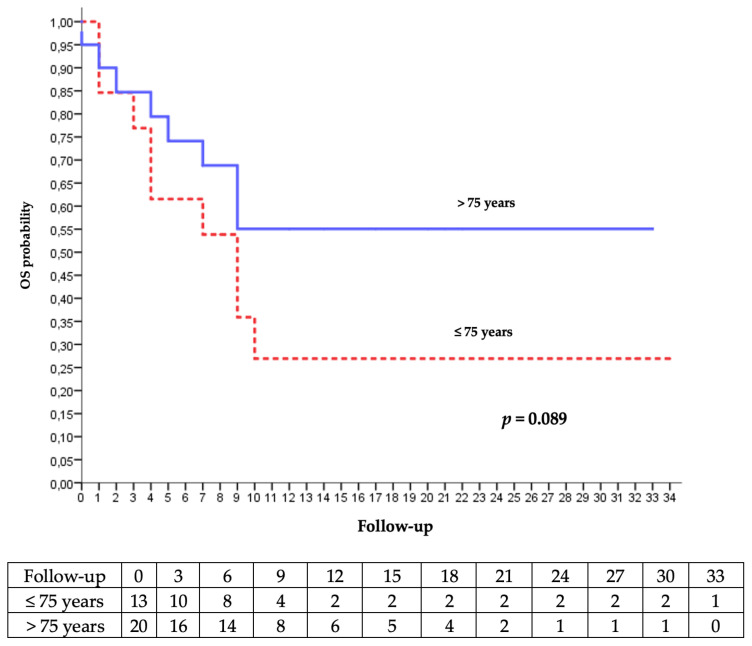
Kaplan–Meier analysis of the occurrence of palliative care decision or death according to age.

**Figure 6 cancers-15-00495-f006:**
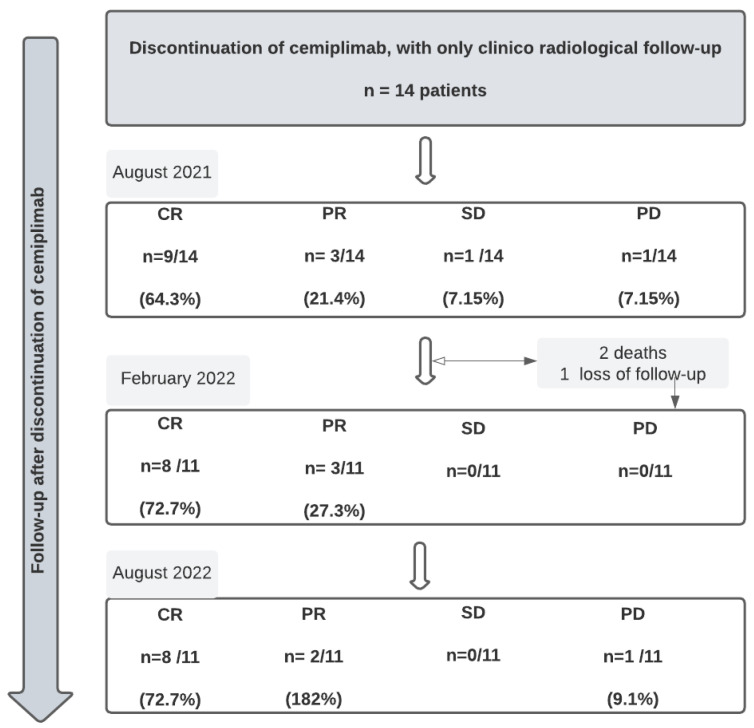
Follow-up of patients after discontinuation of cemiplimab ( CR = complete response; PR = partial response; SD = stable disease; PD = progressive disease).

**Table 1 cancers-15-00495-t001:** Patient characteristics.

	Cemiplimab (*n* = 21)	Cemiplimab/RT (*n* = 12)
Age (years)	75.1 ± 11.8	77.2 ± 12.5
<65 yo	2 (9.5)	3 (25.0)
65–75 yo	5 (23.8)	1 (8.3)
>75 yo	14 (66.7)	8 (66.7)
Gender		
Male	17 (81)	11 (91.7)
Female	4 (19)	1 (8.3)
ECOG status		
0	2 (9.5)	1 (8.4)
1	8 (38.1)	7 (58.3)
2	10 (47.6)	4 (33.3)
3	1 (4.8)	0
Previous cSCC		
No	8 (38.1)	6 (50)
Yes	13 (61.9)	6(50)
Immunodepression		
No	16 (76.2)	8 (66.7)
Yes	5 (23.8)	4 (33.3)
Lymphopenia		
No	14 (66.7)	7 (58.3)
Yes	7 (33.3)	5 (41.7)
Staging		
LacSCC	3 (14.3)	1 (8.3)
mcSCC	18 (85.7)	11 (91.7)
Locoregional metastasis	11 (61.1)	7 (63.6)
Distant metastasis	7 (38.9)	4 (36.4)
Site		
Face	16 (76.2)	6 (50)
Scalp	2 (9.5)	0
Cervical	0	2 (16.7)
Trunk	2 (9.5)	1 (8.3)
Arm or leg	1 (4.8)	3 (25)
Size (mm)	35.4 ± 24.2	48.1 ± 33.4
Previous lines of therapy		
No	19 (90.5)	11 (91.7)
Yes	2 (9.5)	1 (8.3)
Previous radiotherapy		
No	7 (33.3)	11 (91.7)
Yes	14 (66.7)	1 (8.3)
Histological features		
Degree of differentiation		
Well	9 (52.9)	5 (41.7)
Moderate	6 (35.3)	4 (33.3)
Poor	2 (11.8)	3 (25)
PNI		
No	6 (66.7)	5 (100)
Yes	3 (33.3)	0
Bone erosion		
No	10 (66.7)	8 (88.9)
Yes	5 (33.3)	1 (11.1)
Invasion beyond subcutaneous fat		
No	6 (50)	2 (33.3)
Yes	6 (50)	4 (66.7)
Dose of cemiplimab		
3 mg/kg/2 weeks	5 (23.8)	0
350 mg/3 weeks	14 (66.7)	12 (100)
Both #	2 (9.5)	0
Intent of radiotherapy		
Curative		8 (66.7)
Palliative		4 (33.3)
Site of radiotherapy		
Primary tumour		2 (16.7)
Metastasis		10 (83.3)
Dose per fractions (Gy)		4.0 ± 1.7
Fractions		16.2 ± 12.6
Prescribed dose		45.5 ± 22.6
BED		60.5 ± 26.0

Results are expressed as mean ± standard deviation or number (%). The patient characteristics were compared between the two groups. ECOG: Eastern Cooperative Oncology Group; cSCC: cutaneous squamous cell carcinoma; La: locally advanced; m: metastatic; PNI: perineural invasion; Gy: grey; BED: biologically effective dose. #: patients who received the two dosages.

**Table 2 cancers-15-00495-t002:** Best overall responses and disease control rate.

BR	Total		Cemiplimab		Cemiplimab + RT		*p*
	*n*	%	*n*	%	*n*	%	
CR	10	30.3	6	28.6	4	33.3	0.229
PR	5	15.2	4	19.1	1	8.3	
SD	5	15.2	1	4.8	4	33.3	
PD	6	18.2	4	19.0	2	16.7	
NE	7	21.2	6	28.6	1	8.3	
ORR	15	45.5	10	47.6	5	41.6	1.000
DCR	14	70.0	9	81.8	5	55.6	0.336

Results are expressed as number (%), CR = complete response; PR = partial response; SD = stable disease; PD = progressive disease; NE = non evaluated, BR = best response; ORR = objective response rate; DCR = disease control rate; RT = radiotherapy.

**Table 3 cancers-15-00495-t003:** Univariate and multivariate analyses (Cox model).

Variable	Palliative Care Decision or Death		Progression	
	HR (95% IC)	*p*-Value	HR (95% IC)	*p*-Value
Univariate analysis				
Gender M vs. F	0.7 (0.2–2.4)	0.543	0.8 (0.2–3.7)	0.782
Age (years)	0.98 (0.93–1.01)	0.140	0.96 (0.91–1.00)	0.051
65–75 vs. <65	0.6 (0.2–2.5)	0.538	0.3 (0.1–1.4)	0.132
75 vs. <65	0.4 (0.1–1.4)	0.163	0.2 (0.1–0.7	0.014
ECOG status ≥ 2 vs. <2	2.1 (0.9–4.7)	0.076	1.5 (0.6–3.8)	0.378
Previous cSCC	0.8 (0.3–2.2)	0.733	0.5 (0.2–1.7)	0.291
Immunodepression	1.0 (0.4–2.9)	0.969	1.3 (0.4–4.2)	0.713
Lymphopenia < 1 g/L	0.7 (0.3–2.0)	0.515	1.1 (0.4–3.6)	0.830
Staging mcSCC-m vs. LacSCC	1.4 (0.3–5.6)	0.701	2.1 (0.3–16.1)	0.483
Previous lignes of therapy	3.8 (1.1–13.4)	0.041	10.5 (1.9–59.1)	0.008
Degree of differentiation		0.917		0.648
Moderate vs. poor	1.0 (0.2–5.2)	0.995	0.7 (0.2–3.2)	0.672
Well vs. poor	1.3 (0.3–6.1)	0.786	0.5 (0.1–2.3)	0.354
Size of tumour (*n* = 19)	1.01 (0.99–1.03)	0.274	1.02 (1.01–1.05)	0.036
PNI (*n* = 14)	1.0 (0.1–8.6)	0.963	0.7 (0.1–6.2)	0.736
Number of cemiplimab sessions	0.8 (0.7–0.9)	0.001	0.8 (0.7–0.9)	0.005
cemiplimab vs. cemiplimab/RT	0.8 (0.3–2.2)	0.666	1.5 (0.5–4.5)	0.517
Intent of radiotherapy: palliative vs. curative	13.5 (1.5–123.2)	0.021	3.4 (0.7–17.4)	0.144
Multivariate analysis *				
Variable significative	Palliative care decision or death		Progression	
	HR (95% IC)	*p*	HR (95% IC)	*p*
Number of cemiplimab sessions				
Quantitative	0.8 (0.7–0.9)	0.001	0.8 (0.7–10.0)	0.027
Qualitative < 10 vs. ≥10	19.3 (4.2–88.6)	<0.001	-	-

* Variables with *p* < 0.100 in univariate analysis; ECOG: Eastern Cooperative Oncology Group; cSCC: cutaneous squamous cell carcinoma; m: metastatic; La: locally advanced; PNI: perineural invasion; RT: radiotherapy.

## Data Availability

The data presented in this study are available on request from the corresponding author.

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
