# Peer review of "Real-Life Study of the Benefit of Concomitant Radiotherapy with Cemiplimab in Advanced Cutaneous Squamous Cell Carcinoma (cSCC): A Retrospective Cohort Study"

_cancers, 2023, doi:10.3390/cancers15020495_

Round 1

Reviewer 1 Report

Dear authors,

it is a real interesting study. 

Kind regards

Author Response

Dear reviewer,

Best regards,

Barbara BAILLY-CAILLE, MD

Jean-Matthieu L’Orphelin, MD

Reviewer 2 Report

The authors describe a real-life follow-up of cSCC patients treated with cemiplimab with or without additional radiotherapy. Although this is not a clinical trial and therefore results should be interpreted with caution, these data are relevant for daily practice (as there is at the moment nothing else) as well as a basis for designing future trials. 

Some general comments:

(introduction) The references are mainly to guidelines instead of literature investigating for example 'incidence' or 'mortality rate'. This can easily be changed.

(methods) Could the authors clarify about the choices for either initially giving cemiplimab only vs. cemiplimab and radiotherapy. Were there clear-cut criteria for the different treatment choice, was it discussed in a multidisciplinary meeting and decided there? The clearer they can explain why a certain therapy was chosen the more relevant for clinical practice. 

(methods/results) As I said I think it is relevant for clinical practice, but the data should be regarded with caution and comparing the two groups has great risks for interpretation as the inclusion criteria were different. So, in my opinion the focus should be more on showing the data in detail. Even in a supplement per patient, to share the experience as detailed as possible, but not to focus to much on statistical analysis as the groups are small and not comparable. Table 1 can therefore be shown without a p-value. Figure 1 is informative and shows some effect in an early stage, but the end result seems quite comparable. As this is probably a more advanced group, that is supportive for that radiotherapy might be of additional benefit. But that is in my opinion the maximum conclusion we can come to. I have more difficulties with the relevance of figure 2, 3 and table 3. I do not think the data are large enough for building models. Figure 4 can be of assistance to oncologists. (I only would think you would either have < 10 and => 10 and not < 10 or > 9?)

Furthermore, check ',' and '.' for the numbers provided. 

(discussion) Focus more on the subtle conclusions (see comment results) and compare that to other (comparable) data and explain possible differences. Furthermore, try to highlight what this study could mean for designing future trials. 

Author Response

(The authors gave the same response as above.)

Round 2

Reviewer 2 Report

Only minor detail: 

For figure 3 also change the table (everything is changed in 10, there is still a 9)